# Radical Mediated Decarboxylation of Amino Acids via Photochemical Carbonyl Sulfide (COS) Elimination

**DOI:** 10.3390/molecules29071465

**Published:** 2024-03-25

**Authors:** Alby Benny, Lorenzo Di Simo, Lorenzo Guazzelli, Eoin M. Scanlan

**Affiliations:** 1Trinity Biomedical Sciences Institute, Trinity College Dublin, 152-160 Pearse Street, Dublin 2, D02 R590 Dublin, Ireland; bennya@tcd.ie (A.B.); lorenzo.disimo@phd.unipi.it (L.D.S.); 2Department of Pharmacy, University of Pisa, Via Bonanno 33, 56126 Pisa, Italy

**Keywords:** thioacid, decarboxylation, amino acid, radical, photochemistry, blue LED, flow

## Abstract

Herein, we present the first examples of amino acid decarboxylation via photochemically activated carbonyl sulfide (COS) elimination of the corresponding thioacids. This method offers a mild approach for the decarboxylation of amino acids, furnishing *N*-alkyl amino derivatives. The methodology was compatible with amino acids displaying both polar and hydrophobic sidechains and was tolerant towards widely used amino acid-protecting groups. The compatibility of the reaction with continuous-flow conditions demonstrates the scalability of the process.

## 1. Introduction

Since the pioneering study by Barton and co-workers into radical mediated reactions for decarboxylation, these methods have persisted as an essential tool-kit for organic chemists seeking to carry out chemoselective modifications of carboxylic acid residues on organic substrates [1,2]. Recently, the decarboxylation of carboxylic acids under photoredox catalysis has emerged as a mild and facile method for the formation of alkyl radicals which can be trapped to furnish a diverse range of products [3,4,5,6]. MacMillan and co-workers have also reported photoredox-catalyzed decarboxylative arylation of α-amino acids (AAs) in the context of the conversion of biomass into useful pharmacophores [7].

The archetypical thiol-ene reaction furnishes a robust thioether bond between a thiol and an unsaturated residue, and this ‘click’ reaction has been widely exploited for bioconjugation reactions, including peptide and protein modification [8,9]. Recently, we reported the acyl counterpart of the thiol-ene reaction, the acyl thiol-ene (ATE) reaction, as a method to furnish thioesters from the addition of thioacids onto alkenes under radical mediated conditions. Thioacids possess bond-dissociation energies in the same range as alkyl thiols (87 kcal mol^−1^) and radical formation can therefore be initiated under identical conditions [10]. ATE retains the covetable characteristics of the thiol-ene reaction and is believed to proceed via the same reaction mechanism. The ATE reaction has already been exploited by the Scanlan group to synthesize thiolactones via intra- and intermolecular reactions, and to obtain peptidyl/carbohydrate thioesters suitable for S-to-N acyl transfer to form amide bonds [11,12,13].

During our investigations into ATE ligation of amino acids and peptides, COS elimination of the thiyl radical emerged as a competing side reaction to the desired thioester formation for di-substituted thioacid substrates [13]. Presumably due to the instability of primary radical intermediates, COS elimination was not observed as a side reaction for monosubstituted thioacids [13]. To the best of our knowledge, only one previous example of COS elimination under radical mediated conditions exists in the literature, whereby Shimizu and co-workers demonstrated the thermally induced radical dethiocarboxylation of C-terminal thioacid peptides in aqueous buffer using the radical initiator VA-044 to furnish alkylated amide peptides (Figure 1a) [14]. Under aqueous conditions, the authors reported that hydrolysis of the C-terminal thioacid to the corresponding carboxylic acid was a minor but notable side reaction, especially with non-sterically hindered AAs such as glycine and alanine at the C-terminal [14]. A proposed mechanism for the radical mediated COS elimination is depicted in Figure 1b. Following the formation of a thiyl radical via photoinitiation from a suitable thioacid, the elimination of COS forms a carbon-centered radical. Subsequent hydrogen atom transfer (HAT) from the photosensitizer or another thioacid furnishes the dethiocarboxylated product.

Herein, we report the photochemical, radical-mediated dethiocarboxylation of broadly available amino acids via carbonyl sulfide (COS) elimination of the corresponding thioacid derivative, furnishing *N*-alkyl amine products (Figure 1c). This reaction proceeds rapidly under UV and blue LED light irradiation. In addition, the reaction is compatible under continuous flow conditions, demonstrating the potential scalability of the reaction.

## 2. Results and Discussion

### 2.1. Optimization of Dethiocarboxylation Reaction

In order to study the dethiocarboxylation reaction, we required access to thioacid derivatives of AAs. Due to the propensity of thioacids to oxidize and hydrolyze during prolonged storage, we employed a strategy developed by Crich and co-workers whereby the thioacid is protected as an *S*-trityl (STrt) thioester which can be deprotected rapidly with trifluoroacetic acid to furnish the thioacid in quantitative yield after concentration in vacuo [15]. The STrt thioesters are prepared from the corresponding carboxylic acids by coupling with triphenylmethanethiol (TrtSH) under Steglich thioesterification conditions (Figure 1) [15].

To optimize conditions for the dethiocarboxylation reaction, an *N*-fluorenylmethoxycarbonyl (Fmoc) derivative of the amino acid phenylalanine (Phe) was selected as a model substrate. The Fmoc-Phe-STrt thioester **1a** was synthesized from Fmoc-Phe-OH and deprotected to the corresponding thioacid, Fmoc-Phe-SH (**TA-1**), with 25% *v*/*v* TFA/DCM in 5 min. The conversion was monitored by ^13^C NMR spectroscopy to ensure there was quantitative conversion to the desired thioacid (Appendix A) before proceeding with screening conditions for the dethiocarboxylation reaction outlined in Table 1.

High yields of the product, *N*-Fmoc-protected phenethylamine **2a**, formed by the dethiocarboxylation of **TA-1**, were observed under both UV light irradiation (354 nm) using 2,2-dimethoxy-2-phenylacetophenone (DPAP) as a photoinitiator (Table 1, entries 1–2) and blue LED light irradiation (440 nm) with Eosin Y as a photoredox catalyst (Table 1, entries 3–6). Using UV irradiation, an optimal yield of 96% was observed at 15 min reaction time with 0.2 equiv. of DPAP in EtOAc (Table 1, entry 2). The reaction with blue LED light irradiation with 0.25 equiv. of Eosin Y for 1 h in EtOAc gave the best result with a 97% NMR yield (Table 1, entry 4). Having comparable results between the two different light sources, blue LED irradiation was selected to take forward for batch synthesis since it is a milder initial energy source and can be carried out using inexpensive light sources. Moreover, visible light-induced radical reactions benefit from the incapability of most organic molecules to absorb visible light, minimizing possible side reactions and the decomposition of reactants or products by photoactivation. An attempt with a catalytic amount of Eosin Y (1 mol%) gave a satisfactory yield of 72% in 1 h (Table 1, entry 6). A low yield of 6% was obtained without any Eosin Y under blue LED light irradiation (Table 1, entry 7), with the low conversion likely arising from formation of thiyl radicals from molecular O_2_-derived free radicals in solution [16]. When the reaction was conducted in the dark without blue LED irradiation, no reaction was observed (Table 1, entry 8). The radical nature of the dethiocarboxylation reaction was confirmed by the addition of 2 equiv. of TEMPO, which resulted in no reaction (Table 1, entry 9).

### 2.2. Reaction Scope

With the optimized conditions in hand, STrt thioesters **1a**–**1k** were prepared in good yields from commercially available and suitably protected amino acid derivatives (Figure 2). The STrt thioesters were deprotected by treatment with varying concentrations of TFA, depending on whether concomitant deprotection of the N-terminal and side chain-protecting groups were also required to furnish the required thioacids (Section 3.4). The thioacids were concentrated in vacuo following deprotection and used promptly without further purification.

The results of the scope using the optimized conditions are summarized in Figure 3. High isolated yields of the *N*-Fmoc phenethylamine **2a** (72%) and *N*-Boc phenethylamine **2b** (79%) were obtained following dethiocarboxylation, demonstrating the suitability of the method towards the two most common α-amino-protecting groups used for AAs. The free amino derivative **2c** was not isolated. Since the dethiocarboxylation reaction requires a protonated thioacid for proton abstraction to form the thiyl radical, the α-amino group will always be in the protonated alkylammonium form. This may act as an electron withdrawing group that destabilizes the adjacent carbon centered radical that would form upon COS elimination. Similar results have been reported for the thiol-ene click reaction, where proximal α-amino groups have been shown to prevent reaction on peptide substrates [17]. *N*-Ac phenethylamine **2d** was isolated in a poor yield of 27% due to an unidentified side reaction during the deprotection of thioester **1c** to the corresponding thioacid. The glycine derivative **2e** was only detected in trace amounts by ^1^H NMR spectroscopy, with the low rate of COS elimination likely attributable to the high energy primary carbon-centered radical intermediate that is formed. The fully protected serine derivative **2f** was isolated in a moderate yield of 51%. Deprotection of *tert*-butyl ether to the hydroxy group resulted in a significant drop in yield to 16% (**2g**), which increased to 42% with the more substituted threonine derivative **2h**. *N*-alkyl derivatives of proline (**2i**), methionine (**2j**), glutamic acid (**2k**), lysine (**2l**), and tryptophan (**2m**) were isolated in low to moderate yields after dethiocarboxylation of the corresponding thioacids. In the case of **2e**–**2m**, incomplete consumption of the corresponding thioacids were observed by analytical thin-layer chromatography (TLC) after 1 h reaction time under blue LED irradiation, which suggests that increasing reaction times could increase the isolated yield. No obvious general trends were observed for the reactivity of the AAs studied.

### 2.3. Dethiocarboxylation in Flow

In recent years, there has been growing interest in academia and industry in the application of photochemical reactions to continuous flow to increase reaction selectivity, efficiency, and productivity [18,19]. The scaling up of photochemical reactions in batch can suffer from low yields due to poor light penetration, which is easily overcome in flow as the narrow tubing ensures uniform irradiation of the reaction mixture [20]. With this in mind, we set out to investigate whether the photocatalyzed dethiocarboxylation was compatible with flow. The model reaction used for the batch optimization with the dethiocarboxylation of Phe thioacid **TA-1** was applied to flow using UV irradiation since we wanted to minimize reaction time and increase throughput. Gratifyingly, *N*-Fmoc phenethylamine **2a** was isolated in an excellent yield of 86% with 5 min residence time of the reagents under UV irradiation on a 1 mmol scale (Figure 4). This result demonstrates compatibility of the dethiocarboxylation reaction with flow and the potential scalability of the process. 

## 3. Materials and Methods

All commercial chemicals used were supplied by Sigma Aldrich, Burlington, MA, USA (Merck), Fluorochem, VWR Carbosynth, and Tokyo Chemical Industry and used without further purification unless otherwise stated. Deuterated solvents for NMR were purchased from Sigma Aldrich (Merck) or VWR. Solvents for synthesis purposes were used at HPLC grade. All UV reactions were carried out in a Luzchem photoreactor, LZC-EDU (110 V/60 Hz), containing 14 UVA lamps centered at 354 nm. All blue LED reactions were carried out using 2 Kessil PR160-440 LED lamps centered at 440 nm. Silica gel 60 (Merck, 230–400 mesh) was used for silica gel flash chromatography and all compounds were subject to purification using silica gel, unless otherwise stated. Analytical thin layer chromatography (TLC) was carried out with silica gel 60 (fluorescence indicator F254; Merck) and visualized by UV irradiation or molybdenum staining [ammonium molybdate (5.0 g) and concentrated H_2_SO_4_ (5.3 mL) in 100 mL H_2_O]. NMR spectra were recorded using Bruker DPX 400 (400.13 MHz for ^1^H NMR and 100.61 MHz for ^13^C NMR), Bruker AV 400 (400.13 MHz for ^1^H NMR and 100.61 MHz for ^13^C NMR), or Agilent MR400 (400.13 MHz for ^1^H NMR and 100.61 MHz for ^13^C NMR) instruments. Chemical shifts, δ, are in ppm and referenced to the internal solvent signals. NMR data was processed using MestReNova software (version 14.3.3). ESI mass spectra were acquired in positive and negative modes as required, using a Micromass TOF mass spectrometer interfaced to a Waters 2690 HPLC or a Bruker micrOTOF-Q III spectrometer interfaced to a Dionex UltiMate 3000 LC.

### 3.1. General Procedure for the Synthesis of STrt Thioesters 1a–1k (GP-1)

To a solution of carboxylic acid (1 equiv., 0.2 M) in anhydrous DCM under argon was added 1-ethyl-3-(3-dimethylaminopropyl)carbodiimide hydrochloride (1.2 equiv.) and the solution was stirred at 0 °C for 10 min. Then, 4-Dimethylaminopyridine (0.2 equiv.) and triphenylmethanethiol (1 equiv.) were added and the mixture was stirred at room temperature for 18 h. The solvent was evaporated in vacuo and the residue obtained was purified directly by silica gel flash chromatography (*n*-hexane/EtOAc gradient) to afford the desired compound.

#### 3.1.1. S-trityl (*S*)-2-((((9H-fluoren-9-yl)methoxy)carbonyl)amino)-3-phenylpropanethioate (**1a**) [13]

Synthesized from Fmoc-Phe-OH (1.0 g) using GP-1. Foamy white solid (1.46 g, 87%). ^1^H NMR (400 MHz, CDCl_3_) δ 7.75 (d, *J* = 7.7 Hz, 2H), 7.53 (t, *J* = 7.7 Hz 1H), 7.39 (t, *J* = 7.7 Hz 1H), 7.31–7.15 (m, 20H), 7.02 (d, *J* = 7.7 Hz, 2H), 5.09 (d, *J* = 9.0 Hz, 1H), 4.73–4.68 (m, 1H), 4.40–4.32 (m, 2H), 4.19 (t, *J* = 7.2 H, 1H), 3.04–2.93 (m, 2H) ppm. ESI-MS (*m*/*z*) calculated for C_43_H_35_NO_3_S [M+Na]^+^; 668.2229, found 668.2230.

#### 3.1.2. S-trityl (*S*)-2-((tert-butoxycarbonyl)amino)-3-phenylpropanethioate (**1b**) [21]

Synthesized from Boc-Phe-OH (1.0 g) using GP-1. Foamy pale yellow solid (1.55 g, 87%). ^1^H NMR (400 MHz, CDCl_3_) δ 7.29–7.20 (m, 18H), 7.04–6.98 (m, 2H), 4.80 (d, *J* = 8.8 Hz, 1H), 4.60 (m, 1H), 2.97 (dd, *J* = 14.1, 5.9 Hz, 1H), 2.88 (dd, *J* = 14.1, 6.9 Hz, 1H), 1.41 (s, 9H) ppm. ESI-MS (*m*/*z*) calculated for C_33_H_33_NO_3_S [M+Na]^+^; 546.2079, found 546.2081.

#### 3.1.3. S-trityl (*S*)-2-acetamido-3-phenylpropanethioate (**1c**) 

Synthesized from Ac-Phe-OH (1.5 g) using GP-1. Foamy white solid (2.96 g, 88%). ^1^H NMR (400 MHz, CDCl_3_) δ 7.31–7.06 (m, 18H), 6.98–6.85 (m, 2H), 5.60 (d, *J* = 8.5 Hz, 1H), 4.92 (dt, *J* = 8.5, 6.2 Hz, 1H), 3.03–2.82 (m, 2H), 1.85 (s, 3H) ppm. ^13^C {H} NMR (101 MHz, CDCl_3_) δ 196.9, 169.7, 143.4, 135.5, 130.0, 129.7, 128.7, 127.9, 127.3, 127.3, 70.9, 59.5, 38.1, 23.3 ppm. ESI-MS (*m*/*z*) calculated for C_30_H_27_NO_2_S [M+Na]^+^; 488.1663, found 488.1655.

#### 3.1.4. S-trityl 2-((((9H-fluoren-9-yl)methoxy)carbonyl)amino)ethanethioate (**1d**) [13]

Synthesized from Fmoc-Gly-OH (1.0 g) using GP-1. Foamy white solid (1.14 g, 78%). ^1^H NMR (400 MHz, CDCl_3_) δ 7.75 (d, *J* = 7.5 Hz, 2H), 7.57 (d, *J* = 7.5 Hz, 2H), 7.39 (t, *J* = 7.5 Hz, 2H), 7.32–7.20 (m, 17H), 5.27 (t, *J* = 5.7 Hz, 1H), 4.38 (d, *J* = 7.2 Hz, 2H), 4.20 (t, *J* = 7.2 Hz, 1H), 4.12 (d, *J* = 5.7 Hz, 2H) ppm. ESI-MS (*m*/*z*) calculated for C_36_H_29_NO_3_S [M+Na]^+^; 578.1764, found 578.1760.

#### 3.1.5. S-trityl (*S*)-2-((((9H-fluoren-9-yl)methoxy)carbonyl)amino)-3-(tert-butoxy)propanethioate (**1e**) [11]

Synthesized from Fmoc-Ser(O^t^Bu)-OH (1.0 g) GP-1. Foamy white solid (1.19 g, 71%). ^1^H NMR (400 MHz, CDCl_3_) δ 7.80–7.77 (m, 2H), 7.65 (t, *J* = 7.9 Hz, 2H), 7.41 (t, *J* = 7.9 Hz, 2H), 7.32–7.24 (m, 17H), 5.78 (d, *J* = 5.8 Hz), 4.57–4.53 (dd, *J* = 10.0 Hz, *J* = 6.4 Hz, 1H), 4.50–4.46 (m, 1H), 4.37–4.28 (m, 2H), 3.87 (dd, *J* = 8.4 Hz, *J* = 2.2 Hz, 1H), 3.49 (dd, *J* = 8.4 Hz, *J* = 3.2 Hz, 1H), 1.22 (s, 9H) ppm. ESI-MS (*m*/*z*) calculated for C_41_H_39_NO_4_S [M+Na]^+^; 664.2492, found 664.2491.

#### 3.1.6. S-trityl (2*S*,3*S*)-2-((((9H-fluoren-9-yl)methoxy)carbonyl)amino)-3-(tert-butoxy)butanethioate (**1f**) [11]

Synthesized from Fmoc-Thr(O^t^Bu)-OH (1.0 g) GP-1. Foamy white solid (1.35 g, 82%). ^1^H NMR (400 MHz, CDCl_3_) δ 7.76 (dd, *J* = 7.4, 2.5 Hz, 2H), 7.64 (dd, *J* = 10.2, 7.4 Hz, 2H), 7.39 (t, *J* = 7.4 Hz, 2H), 7.32–7.16 (m, 17H), 5.72 (d, *J* = 9.4 Hz, 1H), 4.56 (dd, *J* = 10.2, 6.5 Hz, 1H), 4.41–4.15 (m, 2H), 1.14–1.00 (m, 12H) ppm. ESI-MS (*m*/*z*) calculated for C_42_H_41_NNaO_4_S [M+Na]^+^; 678.2648, found 678.2655.

#### 3.1.7. (9H-fluoren-9-yl)methyl (*S*)-2-((tritylthio)carbonyl)pyrrolidine-1-carboxylate (**1g**) [13]

Synthesized from Fmoc-Pro-OH (1.0 g) using GP-1. Foamy white solid (0.91 g, 51%). ^1^H NMR (400 MHz, CDCl_3_, mixture of cis/trans isomers) δ 7.83–7.77 (m, 2H), 7.70–7.61 (m, 2H), 7.46–7.38 (m, 2H), 7.35–7.18 (m, 17H), 4.60–4.37 (m, 3H), 4.36–4.24 (m, 1H), 3.69–3.69 (m, 1H), 3.59–3.45 (m, 1H), 2.24–2.01 (m, 1H), 2.00–1.79 (m, 3H) ppm. ESI-MS (*m*/*z*) calculated for C_39_H_33_NO_3_S [M+Na]^+^; 618.2079, found 618.2077.

#### 3.1.8. S-trityl (*S*)-2-((((9H-fluoren-9-yl)methoxy)carbonyl)amino)-4-(methylthio)butanethioate (**1h**)

Synthesized from Fmoc-Met-OH (1.0 g) using GP-1. Foamy white solid (1.41, 83%). ^1^H NMR (400 MHz, CDCl_3_) δ 7.79 (d, *J* = 7.6 Hz, 2H), 7.61 (dd, *J* = 7.6, 2.3 Hz, 2H), 7.42 (t, *J* = 7.5 Hz, 2H), 7.34–7.21 (m, 17H), 5.41 (d, *J* = 8.4 Hz, 1H), 4.59 (td, *J* = 8.4, 4.8 Hz), 4.54–4.38 (m, 2H), 4.25 (t, *J* = 7.0 Hz, 1H), 2.35 (t, *J* = 7.0 Hz, 2H), 2.15–2.00 (m, 4H), 1.92–1.79 (m, 1H) ppm. ^13^C {H} NMR (101 MHz, CDCl_3_) δ 197.1, 155.7, 143.3, 141.3, 129.8, 127.8, 127.8, 127.3, 127.1, 125.1, 125.1, 120.0, 70.9, 67.2, 60.1, 47.2, 32.3, 29.6, 15.4 ppm. ESI-MS (*m*/*z*) calculated for C_39_H_35_NO_3_S_2_ [M+Na]^+^; 652.1956, found 692.1951.

#### 3.1.9. Tert-butyl (*S*)-4-((((9H-fluoren-9-yl)methoxy)carbonyl)amino)-5-oxo-5-(tritylthio)pentanoate (**1i**)

Synthesized from Fmoc-Glu(O^t^Bu)-OH (1.0 g) using GP-1. Foamy white solid (1.10 g, 68%). ^1^H NMR (400 MHz, CDCl_3_) δ 7.78 (d, *J* = 7.4 Hz, 2H), 7.61 (dd, *J* = 7.6, 3.0 Hz, 2H), 7.41 (t, *J* = 7.5 Hz, 2H), 7.35–7.21 (m, 17H), 5.58 (d, *J* = 8.2 Hz, 1H), 4.53–4.40 (m, 2H), 4.37–4.29 (m, 1H), 4.25 (t, *J* = 7.1 Hz, 1H), 2.26–1.99 (m, 3H), 1.92–1.76 (m, 1H), 1.46 (s, 9H) ppm. ^13^C {H} NMR (101 MHz, CDCl_3_) δ 197.4, 172.2, 155.7, 143.4, 141.3, 129.8, 127.8, 127.8, 127.2, 127.1, 125.2, 125.1, 120.0, 80.9, 70.7, 67.2, 60.4, 47.1, 31.1, 28.1 ppm. ESI-MS (*m*/*z*) calculated for C_43_H_41_NO_5_S [M+Na]^+^; 706.2603, found 706.2591.

#### 3.1.10. S-trityl (*S*)-2-((((9H-fluoren-9-yl)methoxy)carbonyl)amino)-6-((tert-butoxycarbonyl)amino)hexanethioate (**1j**) [13]

Synthesized from Fmoc-Lys(Boc)-OH (1.0 g) using GP-1. Foamy white solid (1.42 g, 92%). ^1^H NMR (400 MHz, CDCl_3_) δ 7.76 (d, *J* = 7.7 Hz, 2H), 7.59 (d, *J* = 7.4 Hz, 2H), 7.39 (t, *J* = 7.5 Hz, 2H), 7.31–7.18 (m, 17H), 5.33 (d, *J* = 8.2 Hz, 1H), 4.53–4.30 (m, 4H), 4.23 (t, *J* = 7.0 Hz, 1H), 3.13–2.97 (m, 2H), 1.81–1.70 (m, 1H), 1.65–1.53 (m, 1H), 1.49–1.35 (m, 11H), 1.25–1.08 (m, 2H) ppm. ESI-MS (*m*/*z*) calculated for C_45_H_46_N_2_O_5_S [M+Na]^+^; 749.3025, found 749.3021.

#### 3.1.11. Tert-butyl (*S*)-3-(2-((((9H-fluoren-9-yl)methoxy)carbonyl)amino)-3-oxo-3-(tritylthio)propyl)-1H-indole-1-carboxylate (**1k**)

Synthesized from Fmoc-Trp(Boc)-OH (1.0 g) using GP-1. Foamy white solid (1.12 g, 75%). ^1^H NMR (400 MHz, CDCl_3_, mixture of rotamers) δ 8.16 (d, *J* = 8.1 Hz, 1H), 7.77 (d, *J* = 7.5 Hz, 2H), 7.60–7.45 (m, 4H), 7.43–7.33 (m, 3H), 7.30–7.18 (m, 18 H), 5.39 (d, *J* = 8.7 Hz, 1H), 4.94–4.77 (m, 1H), 4.47–4.29 (m, 2H), 4.25–4.11 (m, 1H), 3.19 (dd, *J* = 15.0, 5.8 Hz, 1H), 3.08 (dd, *J* = 15.0, 7.0 Hz, 1H), 1.68 (s, 9H) ppm. ^13^C {H} NMR (101 MHz, CDCl_3_, mixture of rotamers, minor rotamer*) δ 197.2, 155.6, 149.6, 143.8, 143.7, 143.4, 141.3, 135.3, 130.3, 129.8, 127.8, 127.7*, 127.2, 127.1*, 125.2, 125.1*, 124.7, 124.5*, 122.8, 120.0, 119.0, 115.4, 114.6, 83.8, 70.9, 67.4, 60.6, 47.1, 28.2, 27.9* ppm. ESI-MS (*m*/*z*) calculated for C_50_H_44_N_2_O_5_S [M+Na]^+^; 807.2869, found 807.2863.

### 3.2. Dethiocarboxylation Using UV Irradiation for Optimization Table

Thioester **1a** (50 mg) was dissolved in DCM (1 mL) and then TES (20 equiv.) and TFA (25% *v*/*v*) were added and stirred at rt for 5 min under argon. The reaction mixture was concentrated in vacuo and dried to afford the thioacid Fmoc-Phe-SH. Then, Fmoc-Phe-SH (1 equiv., 0.1 M), DPAP (0.2 equiv.) and 1,3,5 trimethoxybenzene (1 equiv.) were dissolved in EtOAc and stirred at room temperature under UV irradiation for 5/15 min. The solvent was removed in vacuo and the sample was analyzed by ^1^H NMR. 

### 3.3. Dethiocarboxylation Using Blue LED Irradiation for Optimization Table

Thioester **1a** (50 mg) was dissolved in DCM (1 mL) and then TES (20 equiv.) and TFA (25% *v*/*v*) were added and stirred at rt for 5 min under argon. The reaction mixture was concentrated in vacuo and dried to afford the thioacid Fmoc-Phe-SH. Then, Fmoc-Phe-SH (1 equiv., 0.1 M), Eosin Y (0.01 or 0.25 equiv.) and 1,3,5 trimethoxybenzene (1 equiv.) were dissolved in EtOAc and stirred at room temperature under blue LED irradiation for 15 min/1 h. The solvent was removed in vacuo and the sample was analyzed by ^1^H NMR. 

### 3.4. General Procedure for Dethiocarboxylation Using Blue LED Irradiation for 2a–2m (**GP-2**)

Thioacid (1 equiv.) and Eosin Y (0.25 equiv.) were dissolved in EtOAc (0.1 M) and stirred at room temperature under blue LED irradiation for 1 h. The solvent was removed in vacuo and the residue obtained was purified by silica gel flash chromatography (*n*-hexane/EtOAc gradient) to afford the desired compound.

#### 3.4.1. (9*H*-fluoren-9-yl)methyl phenethylcarbamate (**2a**) 

Thioester **1a** (400 mg) was dissolved in DCM (0.1 M) and deprotected to the required thioacid with TES (20 equiv.) and TFA (25% *v*/*v*) at rt for 5 min under argon. Following concentration and drying of the reaction mixture in vacuo, the product was synthesized using GP-2. Pale yellow oil (153 mg, 72%). ^1^H NMR (400 MHz, CDCl_3_) δ 7.66 (d, *J* = 7.6 Hz, 2H), 7.67 (d, *J* = 7.6 Hz, 2H), 7.50 (t, *J* = 7.5 Hz, 2H), 7.21 (t, *J* = 7.2 Hz, 4H), 7.08 (d, *J* = 7.5 Hz, 2H), 4.66 (s, 1H), 4.31 (d, *J* = 7.0 Hz, 2H), 4.11 (t, *J* = 7.0 Hz, 1H), 3.36 (q, *J* = 6.8 Hz, 2H), 2.72 (t, *J* = 6.9 Hz, 2H). ^13^C {H} NMR (101 MHz, *CDCl_3_*) δ 156.4, 144.1, 141.5, 138.9, 129.0, 128.8, 127.8, 127.2, 126.6, 125.2, 120.1, 66.7, 47.5, 42.4, 36.3. ESI-MS (*m*/*z*) calculated for C_23_H_21_NO_2_ [M+Na]^+^; 366.1465, found 366.1464.

#### 3.4.2. Tert-butyl phenethylcarbamate (**2b**) 

Thioester **1b** (370 mg) was dissolved in DCM (0.1 M) and deprotected to the required thioacid with TES (20 equiv.) and dropwise addition of TFA (5% *v*/*v*) at 0 °C for 5 min under argon. Following concentration and drying of the reaction mixture in vacuo, the product was synthesized using GP-2. Pale yellow oil (124 mg, 79%). ^1^H NMR (400 MHz, CDCl_3_) δ 7.35–7.19 (m, 5H), 4.57 (bs, 1H), 3.44–3.34 (m, 2H), 2.82 (t, *J* = 7.0 Hz, 2H), 1.46 (s, 9H). ^13^C {H} NMR (101 MHz, CDCl_3_) δ 155.9, 139.0, 128.8, 128.6, 126.4, 41.9, 36.3, 28.4. ESI-MS (*m*/*z*) calculated for C_13_H_19_NO_2_ [M+Na]^+^; 244.1313, found 244.1312.

#### 3.4.3. *N*-phenethylacetamide (**2d**) 

Thioester **1c** (420 mg) was dissolved in DCM (0.1 M) and deprotected to the required thioacid with TES (20 equiv.) and TFA (25% *v*/*v*) at rt for 5 min under argon. Following concentration and drying of the reaction mixture in vacuo, the product was synthesized using GP-2. White solid (40 mg, 27%). ^1^H NMR (400 MHz, CDCl_3_) δ 7.35–7.02 (m, 5H), 5.37 (s, 1H), 3.78–3.26 (m, 2H), 2.75 (t, *J* = 6.9 Hz, 2H), 1.88 (s, 3H). ^13^C {H} NMR (101 MHz, CDCl_3_) δ 170.1, 139.0, 128.8, 126.7, 40.8, 35.8, 23.5. ESI-MS (*m*/*z*) calculated for C_10_H_14_NO [M+H]^+^; 164.1064, found 164.1069.

#### 3.4.4. (9*H*-fluoren-9-yl)methyl (2-(tert-butoxy)ethyl)carbamate (**2f**) 

Thioester **1e** (320 mg) was dissolved in DCM (0.1 M) and deprotected to the required thioacid with TES (20 equiv.) and TFA (25% *v*/*v*) at rt for 5 min under argon. Following concentration and drying of the reaction mixture in vacuo, the product was synthesized using GP-2. Pale yellow oil (87 mg, 51%). ^1^H NMR (400 MHz, CDCl_3_) δ 7.77 (d, *J* = 7.5 Hz, 2H), 7.60 (dd, *J* = 7.5, 1.1 Hz, 2H), 7.44–7.36 (m, 2H), 7.31 (td, *J* = 7.5, 1.2 Hz, 2H), 5.14 (d, *J* = 7.1 Hz, 1H), 4.39 (d, *J* = 7.1 Hz, 2H), 4.24 (t, *J* = 7.1 Hz, 1H), 3.41 (dt, *J* = 33.6, 5.2 Hz, 4H), 1.20 (s, 9H). ^13^C {H} NMR (101 MHz, CDCl_3_) δ 156.6, 144.2, 141.5, 127.8, 127.2, 125.2, 120.1, 73.3, 66.9, 60.8, 47.4, 41.8, 27.7. ESI-MS (*m*/*z*) calculated for C_21_H_25_NO_3_S [M+Na]^+^; 362.1727, found 362.1733.

#### 3.4.5. (9*H*-fluoren-9-yl)methyl (2-hydroxyethyl)carbamate (**2g**) 

Thioester **1e** (375 mg) was dissolved in DCM (0.1 M) and deprotected to the required thioacid with TES (20 equiv.) and TFA (50% *v*/*v*) at rt for 2 h under argon. Following concentration and drying of the reaction mixture in vacuo, the product was synthesized using GP-2. Pale yellow oil (26 mg, 16%). ^1^H NMR (400 MHz, CDCl_3_) δ 7.79 (d, *J* = 7.5 Hz, 2H), 7.59 (d, 2H), 7.47–7.38 (m, 2H), 7.33 (td, *J* = 7.5, 1.2 Hz, 2H), 5.21 (s, 1H), 4.45 (d, *J* = 6.8 Hz, 2H), 4.23 (s, 1H), 3.73 (s, H), 3.37 (d, *J* = 5.3 Hz, 2H), 2.20 (s, 1H), 1.68 (s, 1H). 13 C NMR (101 MHz, CDCl_3_) δ 157.1, 143.9, 141.4, 127.7, 127.1, 125.0, 120.0, 66.8, 62.3, 47.3, 43.5. ESI-MS (*m*/*z*) calculated for C_17_H_17_NO_3_ [M+Na]^+^; 306.1101, found 306.1105.

#### 3.4.6. (9*H*-fluoren-9-yl)methyl (*R*)-(2-hydroxypropyl)carbamate (**2h**) 

Thioester **1f** (440 mg) was dissolved in DCM (0.1 M) and deprotected to the required thioacid with TES (20 equiv.) and TFA (50% *v*/*v*) at rt for 2 h under argon. Following concentration and drying of the reaction mixture in vacuo, the product was synthesized using GP-2. Pale yellow oil (52 mg, 42%). ^1^H NMR (400 MHz, CDCl_3_) δ 7.77 (dt, *J* = 7.6, 1.0 Hz, 2H), 7.59 (dt, *J* = 7.5, 0.9 Hz, 2H), 7.41 (tt, *J* = 7.5, 1.0 Hz, 2H), 7.32 (td, *J* = 7.5, 1.2 Hz, 2H), 5.12 (s, 1H), 4.44 (d, *J* = 6.8 Hz, 2H), 4.22 (t, *J* = 6.8 Hz, 1H), 3.93 (s, 1H), 3.35 (dd, *J* = 14.1, 6.8 Hz, 1H), 3.06 (dt, *J* = 13.7, 6.5 Hz, 1H), 2.00 (s, 1H), 1.19 (d, *J* = 6.3 Hz, 3H). ^13^C {H} NMR (101 MHz, CDCl_3_) δ 157.3, 144.1, 141.5, 127.9, 127.2, 125.2, 120.1, 67.6, 66.9, 48.4, 47.4, 20.9. ESI-MS (*m*/*z*) calculated for C_18_H_19_NO_3_ [M+Na]^+^; 320.1257, found 320.1258.

#### 3.4.7. (9*H*-fluoren-9-yl)methyl pyrrolidine-1-carboxylate (**2i**) 

Thioester **1g** (500 mg) was dissolved in DCM (0.1 M) and deprotected to the required thioacid with TES (20 equiv.) and TFA (25% *v*/*v*) at rt for 5 min under argon. Following concentration and drying of the reaction mixture in vacuo, the product was synthesized using GP-2. Off-white solid (62 mg, 25%). ^1^H NMR (400 MHz, CDCl_3_) δ 7.77 (dd, *J* = 7.5, 1.0 Hz, 2H), 7.63 (dd, *J* = 7.5, 1.0 Hz, 2H), 7.40 (m, 2H), 7.32 (td, *J* = 7.4, 1.2 Hz, 2H), 4.39 (d, *J* = 7.1 Hz, 2H), 4.25 (t, *J* = 7.1 Hz, 1H), 3.48–3.36 (m, 4H), 1.95–1.85 (m, 4H) ppm. ^13^C {H} NMR (101 MHz, CDCl_3_) δ 155.0, 144.2, 141.3, 127.6, 127.0, 125.2, 120.0, 67.1, 47.4, 46.1, 25.4 ppm. ESI-MS (*m*/*z*) calculated for C_19_H_19_NO_2_ [M+Na]^+^; 316.1313, found 316.1312.

#### 3.4.8. (9*H*-fluoren-9-yl)methyl (3-(methylthio)propyl)carbamate (**2j**) 

Thioester **1h** (500 mg) was dissolved in DCM (0.1 M) and deprotected to the required thioacid with TES (20 equiv.) and TFA (25% *v*/*v*) at rt for 5 min under argon. Following concentration and drying of the reaction mixture in vacuo, the product was synthesized using GP-2. Pale yellow oil (83 mg, 32%). ^1^H NMR (400 MHz, DMSO-d_6_) δ 7.89 (d, *J* = 7.4 Hz, 2H), 7.68 (d, *J* = 7.4 Hz, 2H), 7.41 (td, *J* = 7.5, 1.2 Hz, 2H), 7.36–7.28 (m, 3H), 4.31 (d, *J* = 6.8 Hz, 2H), 4.21 (t, *J* = 6.8 Hz, 1H), 3.09–3.01 (m, 2H), 2.43 (t, *J* = 7.3 Hz, 2H), 2.02 (s, 3H), 1.65 (p, *J* = 7.0 Hz, 2H) ppm. ^13^C {H} NMR (101 MHz, DMSO-d_6_) δ 156.1, 143.9, 140.7, 127.6, 127.0, 125.1, 120.1, 65.1, 46.8, 39.2, 30.4, 28.8, 14.6 ppm. ESI-MS (*m*/*z*) calculated for C_19_H_21_NO_2_S [M+H]^+^; 328.1366, found 328.1364.

#### 3.4.9. 4-((((9*H*-fluoren-9-yl)methoxy)carbonyl)amino)butanoic acid (**2k**) 

Thioester **1i** (500 mg) was dissolved in DCM (0.2 M) and deprotected to the required thioacid with TES (20 equiv.) and TFA (90% *v*/*v*) at rt for 3 h under argon. Following concentration and drying of the reaction mixture in vacuo, the product was synthesized using GP-2. Purified by silica gel flash chromatography using 7% MeOH/DCM. White powder (102 mg, 43%). ^1^H NMR (400 MHz, DMSO-d_6_) δ 11.88 (bs, 1H), 7.89 (d, *J* = 7.5 Hz, 2H), 7.68 (d, *J* = 7.6 Hz, 2H), 7.41 (t, *J* = 7.4 Hz, 2H), 7.36–7.28 (m, 3H), 4.29 (d, *J* = 6.9 Hz, 2H), 4.21 (t, *J* = 6.9 Hz, 1H), 3.04–2.81 (m, 2H), 2.24–2.05 (m, 2H), 1.68–1.41 (m, 2H) ppm. ^13^C {H} NMR (101 MHz, DMSO-d_6_) δ 174. 2, 156. 1, 143.9, 140.7, 127.6, 127.0, 125.1, 120.1, 65.2, 46.7, 39.6, 30.9, 24.8 ppm. ESI-MS (*m*/*z*) calculated for C_19_H_19_NO_4_ [M+Na]^+^; 348.1212, found 348.1207.

#### 3.4.10. (9*H*-fluoren-9-yl)methyl tert-butyl pentane-1,5-diyldicarbamate (**2l**) 

Thioester **1j** (500 mg) was dissolved in DCM (0.1 M) and deprotected to the required thioacid with TES (20 equiv.) and TFA (5% *v*/*v*) at 0 °C for 5 min under argon. Following concentration and drying of the reaction mixture in vacuo, the product was synthesized using GP-2. Pale yellow oil (58 mg, 20%). ^1^H NMR (400 MHz, CDCl_3_) δ 7.77 (d, *J* = 7.6 Hz, 2H), 7.69 (d, *J* = 7.5 Hz, 2H), 7.40 (t, *J* = 7.5 Hz, 2H), 7.31 (t, *J* = 7.5 Hz, 2H), 4.79 (bs, 1H), 4.61–4.36 (m, 3H), 4.21 (d, *J* = 7.2 Hz, 1H), 3.24–2.99 (m, 4H), 1.67–1.23 (m, 15H) ppm. ^13^C {H} NMR (101 MHz, CDCl_3_, mixture of rotamers, minor rotamer*) δ 156.5, 156.1, 144.0, 141.3, 127.7, 127.0, 125.1, 120.0, 79.2, 66.5, 47.3, 40.9, 40.3*, 29.8, 29.6*, 28.4, 23.8 ppm. ESI-MS (*m*/*z*) calculated for C_25_H_32_N_2_O_4_ [M+Na]^+^; 447.2260, found 447.2259.

#### 3.4.11. (9*H*-fluoren-9-yl)methyl (2-(1H-indol-3-yl)ethyl)carbamate (**2m**) 

Thioester **1k** (500 mg) was dissolved in DCM (0.1 M) and deprotected to the required thioacid with TES (20 equiv.) and TFA (25% *v*/*v*) at rt for 2 h under argon. Following concentration and drying of the reaction mixture in vacuo, the product was synthesized using GP-2. Pale yellow solid (151 mg, 62%). ^1^H NMR (400 MHz, CDCl_3_) δ 8.04 (bs, 1H), 7.79 (d, *J* = 7.6 Hz, 2H), 7.65 (d, *J* = 7.9 Hz, 1H), 7.59 (d, *J* = 7.5 Hz, 2H), 7.46–7.39 (m, 3H), 7.32 (td, *J* = 7.5, 1.2 Hz, 2H), 7.26–7.22 (m, 1H), 7.18–7.13 (m, 1H), 4.87 (bs, 1H), 4.43 (d, *J* = 7.0 Hz, 2H), 4.23 (t, *J* = 7.0 Hz, 1H), 3.63–3.45 (m, 2H), 3.06–2.84 (m, 2H) ppm. ^13^C {H} NMR (101 MHz, CDCl_3_) δ 156.4, 144.0, 141.3, 136.4, 127.7, 127.1, 127.0, 125.1, 122.3, 122.1, 120.0, 119.6, 118.8, 113.0, 111.2, 66.6, 47.3, 41.3, 25.8 ppm. ESI-MS (*m*/*z*) calculated for C_25_H_22_N_2_O_2_ [M+Na]^+^; 405.1579, found 405.1580.

### 3.5. Dethiocarboxylation in Flow Using UV Irradiation

The flow reactor consisted of a 10 mL syringe pump set to a flow rate of 1.2 mL min^−1^ which ensured a 5 min residence time in the UV reactor. FEP tubing with an inner diameter of 0.8 mm was coiled around a glass insert and placed inside a Luzchem photoreactor, LZC-EDU (110 V/60 Hz), containing 14 UVA lamps centered at 354 nm. The terminus of the tubing was inserted into a glass vial outside the UV reactor for sample collection (Appendix A). Thioacid **TA-1** (1 mmol, 0.1 M) and DPAP (20 mM) dissolved in EtOAc were loaded into the syringe at atmospheric pressure. The flow system was run until the entire reaction volume was pumped through the reactor. The collected sample was concentrated in vacuo and purified by silica gel flash chromatography (1:9 to 1:3 EtOAc:*n*-hexane) to afford compound **2a** as a pale yellow oil (82%).

## Data Availability

The data presented in this study are available on request from the corresponding author.

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
