# Peer review of "Radical Mediated Decarboxylation of Amino Acids via Photochemical Carbonyl Sulfide (COS) Elimination"

_molecules, 2024, doi:10.3390/molecules29071465_

Round 1
Reviewer 1 Report
Comments and Suggestions for Authors
The authors present a novel photoinduced carbonyl sulfide (COS) elimination of thioacids in their manuscript. Under UV or blue LED light irradiation, several significant biomolecules undergo smooth transformation into their corresponding alkyl amino derivatives under mild reaction conditions. Additionally, the authors demonstrate the feasibility of conducting continuous-flow reactions, suggesting promising industrial applicability. While the manuscript shows promise for publication in Molecules, it requires significant revisions addressing the following points:
- Expanded Substrate Scope: The current substrate scope is deemed narrow. To enhance the versatility of the method, the authors are advised to include additional cases illustrating a broader range of transformations.
- Control Experiments for Light Irradiation: The necessity of light irradiation should be substantiated through additional control experiments. Given that previous thermal reactions were conducted at 37°C, demonstrating the indispensability of light irradiation is essential.
- Verification of Hydrogen Source: The authors propose that hydrogen in the desired product originates from the thioacid. However, experimental evidence supporting this claim is lacking. It is essential to provide additional experimental data to validate this conclusion or revise the proposed mechanism accordingly.
- Performance of Specific Substrates: Substrates 2c, 3b, and 4d did not perform satisfactorily under standard reaction conditions, possibly due to their relatively poor absorption of blue light. It is recommended to investigate their performance under UV light irradiation to elucidate any differences in reactivity.
Author Response
Reviewer 1
- Expanded Substrate Scope: The current substrate scope is deemed narrow. To enhance the versatility of the method, the authors are advised to include additional cases illustrating a broader range of transformations.
Six new examples have been included to the scope table including proline, lysine, alanine, tryptophan, glutamic acid and methionine.
- Control Experiments for Light Irradiation: The necessity of light irradiation should be substantiated through additional control experiments. Given that previous thermal reactions were conducted at 37°C, demonstrating the indispensability of light irradiation is essential.
The previous example by Shimuzu and co-workers is still a radical mediated process, the difference being they utilised the thermally induced radical initiator VA-044 to generate the thiyl radical. To demonstrate the indispensability of light irradiation, the reaction was conducted in the dark (Table 1, Entry 8) and no reaction was observed.
- Verification of Hydrogen Source: The authors propose that hydrogen in the desired product originates from the thioacid. However, experimental evidence supporting this claim is lacking. It is essential to provide additional experimental data to validate this conclusion or revise the proposed mechanism accordingly.
The proposed mechanism has been adjusted in Figure 1b since we do not have definitive proof where he proton in abstracted from. Under UV irradiation, it is likely that the proton is abstracted from another thioacid to propagate the radical chain reaction. However, under blue LED conditions using Eosin Y, it is possible that the proton is abstracted from activated Eosin Y.
- Performance of Specific Substrates: Substrates 2c, 3b, and 4d did not perform satisfactorily under standard reaction conditions, possibly due to their relatively poor absorption of blue light. It is recommended to investigate their performance under UV light irradiation to elucidate any differences in reactivity.
The blue light is absorbed by Eosin Y to form the radical that initiates the reaction. Thus, the substrates do no need to absorb blue light. The low yields observed for these compounds are attributed to incomplete consumption of the thioacid after 1 h which has been clarified in the text. To make the results for all substrates comparable, all substrates were subjected to identical reaction conditions.
Reviewer 2 Report
Comments and Suggestions for Authors
Scanlan et. al. reported here an interesting work titled “Radical Mediated Decarboxylation of Amino Acids via Photo-2 catalytic Carbonyl Sulfide (COS) Elimination”. This work was optimized very well from a side reaction and brought an effective and practical method to convert amino acid to the decarboxylated compounds. It is a good complimentary work to previous research reported by themselves. After a number of issues to be addressed, this manuscript is suggested to be accepted.
1) Please renumber the compounds since it looks disordered.
2) 4d was obtained only in 16% yield, what happened? Is there any side reaction occurred? Is there any explanation about the outcome concerning the yield of 4c, 4d, 5b?
3) Some typo error should be checked, for example, the use of “eq. equiv.” should be consistent. Ref 1, 5, 18 should be checked.
4) The abstact should be more relevant to the research.
Comments on the Quality of English LanguageIt's easy to understand.
Author Response
Reviewer 2
- Please renumber the compounds since it looks disordered.
Compounds have been renumbered to be more clear. All thioester are labelled 1a, 1b… and all dethiocarboxylated products have been labelled 2a, 2b…
- 4d was obtained only in 16% yield, what happened? Is there any side reaction occurred? Is there any explanation about the outcome concerning the yield of 4c, 4d, 5b?
The low yields observed for these compounds are attributed to incomplete consumption of the thioacid after 1 h which has been clarified in the text. To make the results for all substrates comparable, all substrates were subjected to identical reaction conditions.
- Some typo error should be checked, for example, the use of “eq. equiv.” should be consistent. Ref 1, 5, 18 should be checked.
All eq. has been changed to equiv. References have also been fully checked.
- The abstact should be more relevant to the research.
Abstract has been modified to more accurately reflect the research content
Reviewer 3 Report
Comments and Suggestions for Authors
Please see the attached document.

English is fine!
Author Response
Reviewer 3
- The scope of the reported transformation is quite limited. Out of 21 AA, only 3 amino acids and its derivatives have been explored. I recommend exploring the scope of the reaction to other amino acids such as lysine, histidine, proline, and glycine.
6 additional examples have been included to the scope table including proline, lysine, alanine, tryptophan, glutamic acid and methionine.
- It would be also great to see if this protocol works with peptides (at least 3 or 4-mer). Authors claim reaction works with important biomolecules, so it is important to show that reaction works with biologically relevant compounds like peptides.
In reference 14, Shimizu and co-workers, demonstrated the thermally induced radical dethiocarboxylation of C-terminal thioacid peptides in aqueous buffer using the radical initiator VA-044 to furnish alkylated amide peptides. Although we have not demonstrated our methodology on peptides, we are confident that the work by Shimizu provides good evidence that it can be applied to biomolecules. This will form part of our future work on this topic.
- Recently, the decarboxylation of carboxylic acids under photoredox catalysis has emerged as a mild and facile method for the formation of alkyl radicals which can be trapped to furnish a diverse range of product[3,4]......... Author should cite additional references for traditional as well as photoredox mediated of decarboxylation amino acid for examples.
Additional references have been citied [4,5]
- Page 5, Line 148, The model reaction used for the batch optimisation with the dethiocarboxylation of Phe thioacid 1a was applied to flow using UV irradiation.... Did author try blue LED for flow chemistry. Any information on such experiment would help reader to understand the difference of UV vs Blue LED for flow chemistry.
Due to the nature of our experimental set-up, we were unable to perform the blue LED reaction in flow. This is something we are working on developing for future applications.
- There are several typos in the manuscript. For examples, Page 2, line 32....possess bond [disassociation] energies.......... Page 7 line 226-227 ....To a stirred solution of S-trityl thioesters 1-5 (1 equiv.) in DCM under argon was added triethylsylane (20 equiv.).
Typos have been corrected.
Round 2
Reviewer 1 Report
Comments and Suggestions for Authors
Upon review, it is evident that the authors have made concerted efforts to address the prevailing concerns. However, it is imperative to refrain from outright denial of straightforward inquiries pertaining to the mechanism and reaction issues. Instead, a more effective approach would involve implementing experimental interventions to navigate these existing challenges. Regarding the assertion concerning light absorption, I respectfully disagree with the authors' standpoint. Light absorption transcends the scope of the photosensitizer alone; it encompasses the entirety of the reaction system. Furthermore, the light absorption characteristics of the photosensitizer are subject to influence by both reactants and solvents. Consequently, I propose that the authors consider conducting additional control experiments to effectively address the current concerns.
Author Response
We disagree with the reviewers comments and will not be conducting any additional experiments. It is up to the editor to decide if they wish to accept the manuscript for publication or not or else we will publish in a different journal.
Reviewer 3 Report
Comments and Suggestions for Authors
All the comments have been successfully addressed. No more comments.
Author Response
Thank you, no comments to address